

# Evolution of sexual dimorphism and Rensch's rule in the beetle genus *Limnebius* (Hydraenidae): is sexual selection opportunistic?

Andrey Rudoy and Ignacio Ribera

Institute of Evolutionary Biology, CSIC-Universitat Pompeu Fabra, Barcelona, Spain

## ABSTRACT

Sexual size dimorphism (SSD) is widespread among animals, with larger females usually attributed to an optimization of resources in reproduction and larger males to sexual selection. A general pattern in the evolution of SSD is Rensch's rule, which states that SSD increases with body size in species with larger males but decreases when females are larger. We studied the evolution of SSD in the genus *Limnebius* (Coleoptera, Hydraenidae), measuring SSD and male genital size and complexity of ca. 80% of its 150 species and reconstructing its evolution in a molecular phylogeny with 71 species. We found strong support for a higher evolutionary lability of male body size, which had an overall positive allometry with respect to females and higher evolutionary rates measured over the individual branches of the phylogeny. Increases in SSD were associated to increases in body size, but there were some exceptions with an increase associated to changes in only one sex. Secondary sexual characters (SSC) in the external morphology of males appeared several times independently, generally on species that had already increased their size. There was an overall significant correlation between SSD, male body size and male genital size and complexity, although some lineages with complex genitalia had low SSD, and some small species with complex genitalia had no SSD. Our results suggest that the origin of the higher evolutionary variance of male body size may be due to lack of constraints rather than to sexual selection, that may start to act in species with already larger males due to random variation.

## INTRODUCTION

Sexual size dimorphism (SSD) is highly variable among animal species, from minuscule males with comparatively giant females to males much larger than females (*Darwin, 1871*; *Hedrick & Temeles, 1989*; *Fairbairn, 1997*; *Vollrath, 1998*). In most insect species, females are larger than males (*Darwin, 1871*; *Arak, 1988*; *Shine, 1988*; *Fairbairn, 1997*), a fact usually explained because the energetic investment in the progeny is larger in females than in males, which mostly provide just genetic information. For this reason, females should be as big and males as small as possible, to minimise resources spend on their

Corresponding author
Ignacio Ribera,
ignacio.ribera@ibe.upf-csic.es

maintenance (*Darwin, 1871*; *Thornhill & Alcock, 1983*; but see *Shine, 1988* for some alternative views).

Although some species have reached this state of minimised males, in the absence of selection both sexes will tend to have the same size, given the strong genetic correlation between sexes for most traits (*Lande, 1980*). However, rapid changes in SSD can occur even when selection pressure is small (*Reeve & Fairbairn, 2001*), which begs the question not why sexual dimorphism exists, but why there are so many species in which males are about the same size or bigger than females. One reason could be ecological (*Slatkin, 1984*; *Shine, 1989*; *Mysterud, 2000*): when sexes have widely different sizes, they may not be able to share the same ecological niche. This may be an advantage in avoiding intraspecific competition, but there is little evidence that SSD may have originated primarily through ecological divergence in any group (*Fairbairn, 1997*). In other species, males contribute to raise the progeny either by protecting females or providing resources, thus equalizing the investment of the two sexes (*Andersson, 1994*). An alternative explanation is sexual selection: larger males may have an advantage, either because they can gain better access to females (male–male competition), or because females prefer them (female choice). In the first case, differences may affect the size of the body, but also other structures used in male–male competition. Similarly, in the latter case, in addition to male body size other characters may be involved, especially genital characters when there is cryptic female choice (*Eberhard, 1985*; *Kuijper, Pen & Weissing, 2012*).

Despite the large body of work on sexual dimorphism, there is still a lack of understanding of its long-term evolution in diverse lineages, particularly among invertebrates (*Fairbairn, 1997*). There are several unresolved questions on the origin and evolution of sexual dimorphism that can be addressed with a phylogenetic reconstruction in speciose lineages with a diversity of male genital and body sizes. One of the few recognised general trends in the evolution of SSD is the so-called Rensch's rule (*Abouheif & Fairbairn, 1997*; *Fairbairn, 1997*; *Székely, Freckleton & Reynolds, 2004*). *Rensch (1950)* observed that sexual differences increased with body size in species in which males were larger, but decreased in species in which females were larger. This implies that male body size varies more over evolutionary time than female body size, irrespective of which sex is larger (*Fairbairn, 1997*). *Fairbairn & Preziosi (1994)* hypothesized that sexual selection for large male size may be the primary force driving Rensch's rule. The observation of Rensch's rule in characters subjected to sexual selection but likely not to other types of selection, such as male wing pigmentation (*Santos & Machado, 2016*), supports this interpretation. An alternative possibility is that males have a larger evolutionary plasticity, somehow equivalent to a larger intraspecific phenotypic plasticity (*Fairbairn, 2005*; *Gómez-Mestre & Jovani, 2013*).

Data supporting Rensch's rule is mostly intraspecific or from closely related small species groups (*Abouheif & Fairbairn, 1997*; *Fairbairn, 1997*; *Székely, Reynolds & Figuerola, 2000*; *Kraushaar & Blanckenhorn, 2002*), with only a few global studies of diverse lineages, and mostly among vertebrates (e.g., *Lindenfors, Székely & Reynolds, 2003*; *Székely, Freckleton & Reynolds, 2004*). In two recent reviews, *Blanckenhorn et al. (2007)* and

*Blanckenhorn, Meier & Teder (2007)* found strong support for Rensch's rule in some groups of Arthropods (some Diptera and Hemiptera Gerridae), with negative allometry in plots of females (*y*-axis) vs. males (*x*-axis); while in others there was isometry (e.g., some beetles and hymenopterans) or only weak trends (butterflies and spiders). Rensch's rule was mostly supported in groups with males larger than females, something unusual in ectotherms (contrary to mammals and birds, *Fairbairn, 1997*), and there was little evidence to support its prevalence at the intraspecific level (see also *Martin et al., 2017*).

In this work, we reconstruct the macroevolutionary patterns of SSD evolution in a diverse lineage of insects with the aim to investigate the origin and evolution of body size differences. We aim to determine the underlying causes of Rensch's rule over long evolutionary periods, and in particular whether it is driven by sexual selection or not. We focus on the relationship between SSD and evolutionary changes in body size of males and females, and whether SSD is linked to size variation in both sexes or can appear through changes in one sex only (*Fairbairn, 1997*). We also study the correlation of SSD with other characters of the male genitalia usually assumed to be the result of sexual selection, such as size and complexity.

As a study group, we use a diverse and ancient lineage of beetles, the genus *Limnebius* (family Hydraenidae). *Limnebius* includes ca. 150 species with an almost cosmopolitan distribution, all of them aquatic, living in all types of continental waters with the only exception being the absence from saline habitats (*Perkins, 1980*; *Jäch, 1993*; *Hansen, 1998*; *Rudoy, Beutel & Ribera, 2016*). In a recent work, *Limnebius* was shown to be divided in two sister lineages with an estimated Oligocene origin, the subgenera *Bilimneus* and *Limnebius* s.str., with ca. 60 and 90 described species respectively (*Rudoy, Beutel & Ribera, 2016*). The two subgenera differ in a number of traits, including variation in body size and in the size and complexity of the male genitalia, much larger in *Limnebius* s.str. (*Rudoy, Beutel & Ribera, 2016*; *Rudoy & Ribera, 2016*). They also differ in sexual dimorphism and the presence of secondary sexual characters (SSC) in the external morphology. In *Bilimneus*, females are slightly larger than males, which do not have strongly developed SSC; on the contrary, within *Limnebius* s.str., there is a wide range of different situations, including males much larger than females and with well-developed SSC (*Jäch, 1993*; *Rudoy & Ribera, 2016*), thus providing a suitable system for the study of the origin and evolution of SSD.

## MATERIALS AND METHODS

### Taxon sampling

We obtained morphological data of the males of 120 and the females of 86 of the ca. 150 described species of *Limnebius*, among them four undescribed species (Table S1). Females were identified mostly by association with males, as there are few characters that could identify them unequivocally (*Perkins, 1980*; *Jäch, 1993*). In the cases in which an unambiguous identification was not possible (for example, when several species of similar size and no other morphological difference coexist in the same locality) females were discarded, resulting in some species for which only males could be studied (Table S1). For some species, only a limited number of specimens could be studied (e.g., *Limnebius*

*paranuristanus*, *Limnebius angustulus* or *Limnebius fontinalis*; see Table S1 for the taxonomic classification of the genus). In most cases, the SSD of these species was very similar to that of their closest relatives, and only in a few instances (e.g., *Limnebius canariensis*) the specimens that could be obtained differed in SSD from related species, raising the possibility of a sampling bias.

## Morphometric measurements

We measured body length of adults (males and females) as the sum of the individual maximum lengths of pronotum and elytra, as the different position of the articulation between the two could alter the total length when measured together. Similarly, the head was not measured, as in many specimens it was partly concealed below the pronotum. Measures were obtained with stereoscope microscopes equipped with an ocular micrometer. In most analyses, we estimated SSD as the direct difference between male and female body sizes, measured in millimetres. In some analyses, we also used the ratio male body size/female body size (rSSD).

Measures of the genitalia were obtained from *Rudoy & Ribera (2016)*. Briefly, male genitalia (aedeagi) were dissected and mounted on transparent labels with dimethyl hydantoin formaldehyde. For size measurements, we used as a single value the average of each measure in all studied specimens of the same species (Table S1). For shape characterisation, a single specimen was used as species show in general a very constant shape of the aedeagus, with very low intraspecific variability as compared with the marked differences between species (*Jäch, 1993*; *Rudoy, Beutel & Ribera, 2016*). We measured the maximum length of the male genitalia orientated in ventral view according to the foramen. We did not include setae or apical membranous structures but included appendages when they were longer than the median lobe (as in e.g., some species of the *Limnebius nitidus* group, *Rudoy, Beutel & Ribera, 2016*). Measurements were directly obtained from the digital images using ImageJ v.1.49 (National Institutes of Health, Bethesda, MD, US, http://imagej.nih.gov/ij/) (Fig. S1).

We used two different measures to characterise the complexity of the aedeagus, following *Rudoy & Ribera (2016)*:

(1) Perimeter of the aedeagus in ventral view, including the median lobe and the main appendages. We obtained an outline of the genitalia from digital images using ImageJ. The total perimeter was the sum of the values of the different parts of the genitalia (median lobe and left parameter, plus main appendages if present, see *Rudoy, Beutel & Ribera, 2016*). We standardised the values by dividing the perimeter by the length of the aedeagus, to obtain a measure of complexity by unit of length (Fig. S1; Table S1).

(2) Fractal dimension. The fractal dimension is a measure of the complexity of a bidimensional projection of the shape of the genitalia, and we use it in addition to the perimeter as a complementary measure of complexity. Given the intricate three-dimensional structure of the male genitalia of some species of *Limnebius* (see *Rudoy, Beutel & Ribera, 2016* for details), we opted for different measures to try to avoid biases due to the unavoidable simplification of reducing these complex structures to a

unidimensional statistic. We estimated the fractal dimension of the outline of the aedeagus in ventral view on images of standard size (2,100 × 2,100 pixels, 2,000 pixels from base to apex of the aedeagus) with the software Fractal Dimension Estimator (http://www.fractal-lab.org/index.html). This software estimates the Minkowski fractal dimension of bidimensional images using the box-counting method (*Falconer, 1990*). The software converts the image to binary data, selects the scaling window of the box, and counts how many boxes are necessary to cover the image. The absolute value of the slope of a log–log graph of the scale with the number of boxes is the fractal dimension of the image (Fig. S1; Table S1).

## Phylogenetic analyses

For our analyses, we use two phylogenetic reconstructions, one with the species for which there were molecular data available and a second one with all species for which there were morphological data, interspersed in the molecular phylogeny according to their morphological similarities.

The molecular phylogeny was based on that obtained in *Rudoy, Beutel & Ribera (2016)* and *Rudoy & Ribera (2016)*, including 71 species of *Limnebius* (Table S2). Taxon sampling was denser for the Palaearctic lineages in subgenus *Limnebius* s.str., including the full range of body sizes and structural variation of the aedeagus. We used as outgroup and to root the tree the genus *Laeliaena*, considered to be sister to *Limnebius* based on multiple morphological synapomorphies (*Hansen, 1991*; *Jäch, 1995*; *Perkins, 1997*; *Beutel, Anton & Jäch, 2003*).

The phylogeny was constructed with Bayesian methods in BEAST 1.8 (*Drummond et al., 2012*) using a combined data matrix with three partitions, (1) mitochondrial protein coding genes (two *cox1* fragments plus *nad1*); (2) mitochondrial ribosomal genes (*rrnL* plus *trnL*); and (3) nuclear ribosomal genes (*SSU* plus *LSU*) (Table S2; *Rudoy & Ribera, 2016*), with a Yule speciation process as the tree prior and an uncorrelated relaxed clock.

Trees were calibrated with the rates estimated in *Cieslak, Fresneda & Ribera (2014)* for family Leiodidae, within the same superfamily Staphylinoidea (*Beutel & Leschen, 2005*) and the same gene combination based on the tectonic separation of the Sardinian plate 33 Ma. It must be noted that for our objectives only relative rates are needed. An absolute calibration would only be necessary to obtain absolute estimates of character change, which is not our main objective and does not affect our conclusions.

We reconstructed the ancestral values of the morphological traits from the values of the terminals (extant species) in BEAST 1.8. We implemented a Brownian motion (BM) model of evolution, a null model of homogeneous evolution in which variation accumulates proportionally with time, with incremental changes drawn from a random distribution with zero mean and finite constant variance (*Hunt & Rabosky, 2014*; *Adams, 2014*). The reconstruction of ancestral values using a BM model of evolution is biased towards average or intermediate values (*Pagel, 1999*; *Finarelli & Goswami, 2013*), which may result in an underestimation of the rates of evolution of some characters. Due to these

limitations, our reconstruction needs to be understood as the simplest null model explaining the evolutionary change in the studied characters.

We reconstructed the origin and secondary loss of some SSC with MESQUITE v.3 (*Maddison & Maddison, 2015*) using parsimony. Secondary male sexual characters in the external morphology of species of *Limnebius* affect mostly the tibiae and the last abdominal sternites (*Jäch, 1993*). In many species of *Limnebius* (but mostly in *Limnebius* s. str.), males have slightly curved and apically wider tibiae, especially in the anterior and medial legs, which also have suction setae (*Jäch, 1993*). However, these characters are difficult to quantify precisely and more observations are needed to establish their prevalence. The hind tibiae of males are also modified in some species of *Limnebius* s.str. (Fig. S2). There are three different types of SSC in the abdominal sternites of males: (1) a more or less developed medial protuberance; (2) two parallel tuffs of setae; and (3) a medial impression delimited by ridges (*Jäch, 1993*; Fig. S2). All of them occur mostly in large species.

We studied the evolution of the morphological characters trough the full evolutionary path of species (i.e., from root to tips) and in the individual branches, using phylogenetic ancestor–descendant comparisons (*Baker et al., 2015*; *Rudoy & Ribera, 2016*). We measured three values for each of the individual branches (including terminals): (1) amount of phenotypic change, equal to the arithmetic difference between the final and initial values of the branch; (2) absolute amount of phenotypic change, equal to the absolute value of the amount of phenotypic change; and (3) phenotypic change measured in darwins (*Haldane, 1949*), computed as the absolute value of the natural logarithm of the ratio between the final and initial values divided by the length of the branch in million years (Table S3). The use of the natural logarithm standardises the change so it is proportional and directly comparable among species with different sizes (*Haldane, 1949*; *Gingerich, 2009*). To qualitatively characterise phenotypic change in the individual branches, we coded as positive or negative the increase or decrease of body size in each sex, as well as the SSD measured as the absolute difference between male and female body size. An estimated change lower than 5% in body size of males or females was considered within experimental error (i.e., 'without change'). For SSD, we considered as 'without change' branches with a change lower than 5% of the total range of observed differences. As individual branches are in principle independent from each other, we analysed these variables with standard statistical procedures (see e.g., *Baker et al., 2015*).

We estimated the phylogenetic signal of the morphological variables in the whole tree using the $K$ metric (*Blomberg, Garland & Ives, 2003*), which tests whether the topology and branch lengths of a given tree better fits a set of tip data compared with the fit obtained when the data have been randomly permuted. The higher the $K$ statistic, the more phylogenetic signal in a trait. $K$ values of 1 correspond to a BM model, which implies some degree of phylogenetic signal. $K$ values closer to 0 correspond to a random or convergent pattern of evolution, while $K$ values greater than 1 indicate strong phylogenetic signal. We used the R package 'Picante' (*Kembel et al., 2010*) to compute $K$ and the significance test. We also measured the correlation between some variables across the whole tree with a regression of phylogenetic independent contrasts with the PDAP

package in MESQUITE v.3. We use a type II regression with reduced major axis to relate the independent contrasts obtained in PDAP of log10 male and female size (*Fairbairn, 1997*; *Blanckenhorn, Meier & Teder, 2007*) using the package PAST v.3 (*Hammer, Harper & Ryan, 2001*).

To test for the possible effect of the incomplete taxon sampling, in some analyses we also used a phylogeny including species with only morphological data. We used the tree provided in *Rudoy, Beutel & Ribera (2016)*, in which species without molecular data were placed mostly according to the similarities of the male aedeagus (Fig. S3). When relationships were uncertain, a polytomy was formed with all the species sharing a similar structure of the aedeagus, and whenever necessary for the analyses these polytomies were randomly resolved in MESQUITE v.3. For comparison, some of the correlations were also repeated using the species values directly, without phylogenetic correction.

## RESULTS

### Overall interspecific allometry in SSD

Both for the whole genus *Limnebius* and subgenus *Limnebius* s.str., the slopes of the regression between the size of females and the size of males were always significantly larger than one (Fig. 1; Table 1), i.e., there was a positive allometry in the size of males with respect to females. Although the estimate of the slope of the regression for subgenus *Bilimneus* was also significantly larger than one when using the estimated molecular phylogeny, the 95% confidence interval when using the phylogeny with all species or the raw data could not reject isometry between both sexes (Table 1).

The phylogenetic signal $K$ of the SSD, as measured with the ratio male/female body size (rSSD), was lower than one, suggesting lack of phylogenetic signal ($K = 0.56$; $p < 0.001$). The $K$ values for body size of males and females were, on the contrary, clearly larger than one, suggesting a strong phylogenetic signal (1.35 and 1.65, respectively, both $p < 0.01$).

For the whole genus *Limnebius*, rSSD was significantly correlated with male and female body size, but correlation values were much larger for male than for female body sizes (Table 2). When the two subgenera were considered separately, the correlation with female body size was not significant in subgenus *Limnebius* s.str., when using the phylogeny with all species included, and never in *Bilimneus* (Table 2).

The correlations of rSSD with the measures of aedeagus size and complexity were also highly significant, although weaker when using the phylogenetic tree with all species than when using the phylogeny with only species with molecular data, or when using raw data without phylogenetic correction (Table 3). Correlations were also weaker within the species of *Bilimneus*, and significant only for the perimeter.

### Evolution of SSD

According to our reconstruction, the ancestral condition of *Limnebius* was a SSD close to a 1:1 ratio (Fig. 2; Table S3). The general evolution in *Bilimneus* was to a slight decrease in size, with females larger than males, while in *Limnebius* s.str., the general trend was

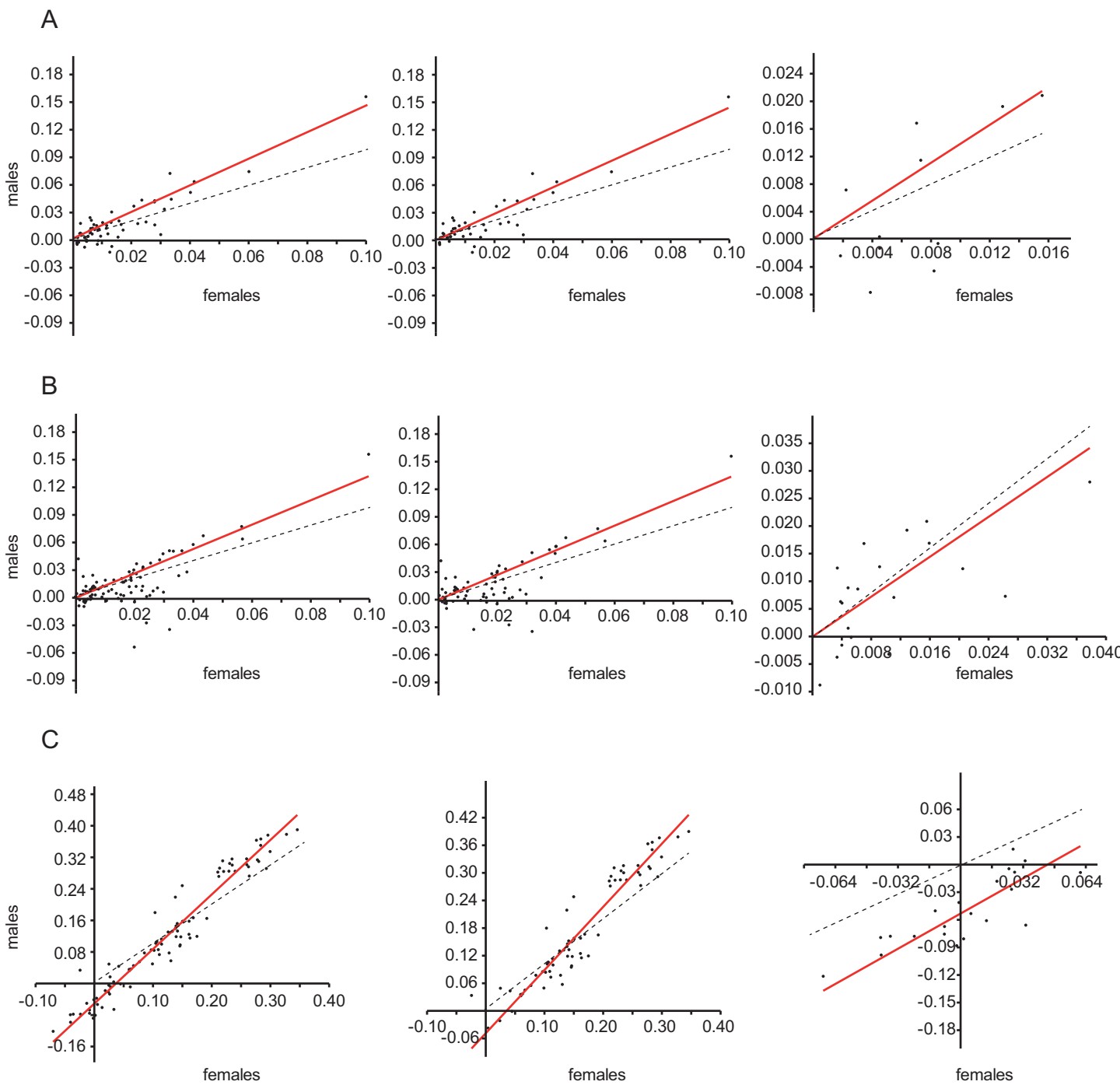

**Figure 1 Regression between male and female body size.** Reduced major axis (RMA) regression between male (*y*-axis) and female (*x*-axis) body size. (A) Phylogenetic independent contrasts of species in the molecular phylogeny (see Figs. 2 and 5); (B) Phylogenetic independent contrasts of the estimated phylogeny with all species (Fig. S3); (C) Regression with the raw data (without phylogenetic correction). Left column, genus *Limnebius*; central column, subgenus *Limnebius* s.str.; right column, subgenus *Bilimneus*. Dotted lines, isometric relationship (slope = 1). See Table 1 for the numerical values of the regressions.

an increase in size (although some lineages maintained the ancestral small size), with males larger than females (Fig. 2; Table S3). This increase in size and SSD was continuous for the extant species with the largest SSD.

**Table 1 Regression between male and female body size.** Type II regression (MRA) between male and female body size. Regressions through the origin using a phylogenetic correction (molecular and including all species, Figs. 2, 5 and S3) were done using PDAP contrasts.

|  | Phylogeny | $n$ | Slope | 95% Interval | $R^2$ | $p$ |
|---|---|---|---|---|---|---|
| *Limnebius* | Molecular | 59 | 1.44 | [1.32–1.63] | 0.82 | <0.0001 |
| *Limnebius* s.str. | Molecular | 47 | 1.44 | [1.31–1.66] | 0.82 | <0.0001 |
| *Bilimneus* | Molecular | 12 | 1.38 | [1.04–3.30] | 0.46 | <0.05 |
| *Limnebius* | All species | 89 | 1.32 | [1.20–1.51] | 0.52 | <0.0001 |
| *Limnebius* s.str. | All species | 68 | 1.34 | [1.22–1.52] | 0.58 | <0.0001 |
| *Bilimneus* | All species | 21 | 0.90 | [0.51–1.10] | 0.42 | <0.001 |
| *Limnebius* | Raw data | 89 | 1.38 | [1.31–1.45] | 0.94 | <0.0001 |
| *Limnebius* s.str. | Raw data | 68 | 1.37 | [1.26–1.48] | 0.89 | <0.0001 |
| *Bilimneus* | Raw data | 21 | 1.20 | [0.82–1.45] | 0.67 | n.s. |

**Table 2 Correlation of rSSD with male and female length.** Correlation ($R^2$) of rSSD with male and female length.

|  | *Limnebius* | | *Limnebius* s.str. | | *Bilimneus* | |
|---|---|---|---|---|---|---|
| **Phylogeny** | $m$ | $f$ | $m$ | $f$ | $m$ | $f$ |
| Molecular tree | 0.63*** | 0.32*** | 0.64*** | 0.33*** | 0.49* | n.s. |
| ALL species | 0.41*** | 0.06* | 0.42*** | n.s. | n.s. | n.s. |
| Raw data | 0.72*** | 0.48*** | 0.68*** | 0.43*** | 0.72*** | n.s. |

Notes:
* $p < 0.05$;
** $p < 0.01$;
*** $p < 0.001$.

**Table 3 Correlation of rSSD with genital traits.** Correlation ($R^2$) of rSSD with aedeagus length ($lg$), perimeter ($per$), and fractal dimension ($fd$).

| **Phylogeny** | $lg$ | $per$ | $fd$ |
|---|---|---|---|
| Molecular tree | 0.13** | 0.40*** | 0.17*** |
| All species | 0.06* | 0.19*** | 0.08** |
| Raw data | 0.46*** | 0.46*** | 0.52*** |

Notes:
* $p < 0.05$;
** $p < 0.01$;
*** $p < 0.001$.

Of the 130 individual branches of the phylogenetic molecular tree, only in 32 did the reconstructed female body size have a faster evolutionary rate than the male body size, as measured in darwins (average difference of male minus female darwins = 0.006, std = 0.017; Fig. 3). Many of the branches in which females evolved faster than males were in the *Limnebius nitidus* subgroup, with an uncertain basal topology but reconstructed as having an overall decrease in SSD (Figs. 2, 4F and S5). Differences were similar when measured with absolute phenotypic change, with only 33 branches out of 130 in which female body size changed more than male body size (average difference of male minus female absolute body size = 0.029 mm, std = 0.045) (Figs. 4D and S5; Table S3).

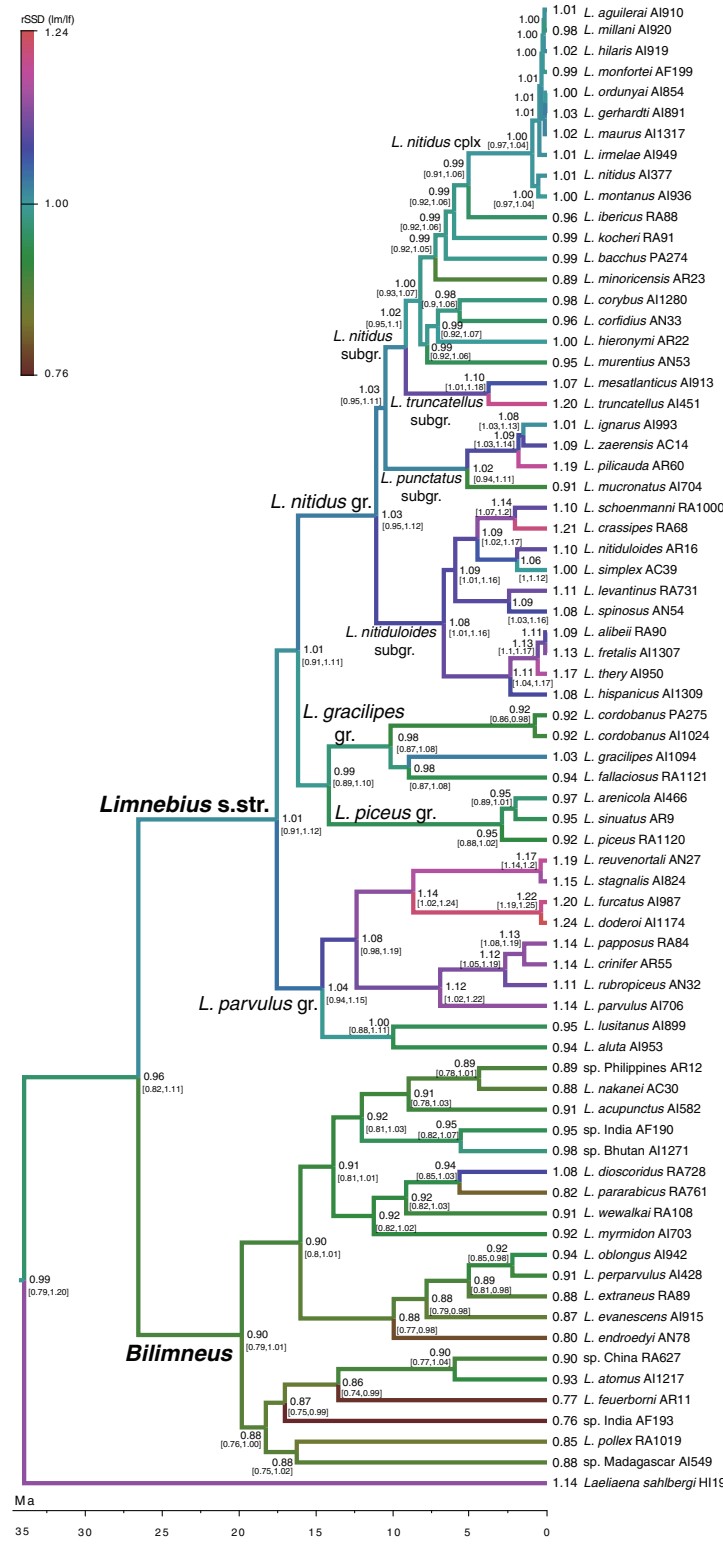

**Figure 2  Evolution of SSD in *Limnebius*.** Evolution of the sexual dimorphism (rSSD) in the phylogeny of *Limnebius*, as reconstructed in BEAST using a Brownian motion model of evolution. Numbers in nodes, reconstructed value of rSSD with 95% confidence interval in square brackets (see Fig. S4 for node support values).

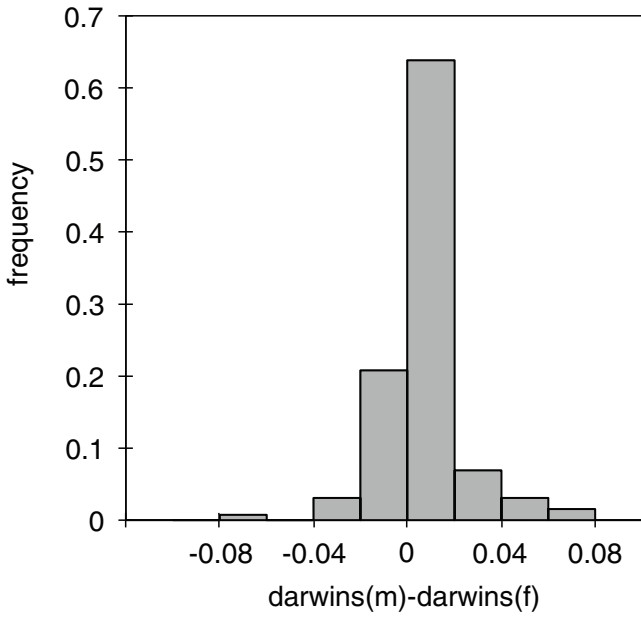

**Figure 3 Differences in phenotypic change between males and females.** Histogram of the differences in darwins of the phenotypic change between males and females in the individual branches of the phylogeny (see Table S3 for the values of the individual branches).

The reconstructed changes in SSD in the individual branches of the phylogenetic tree were clearly associated to overall changes in body size. In almost 50% of the branches (63 out of 130) SSD increased when the body size of males and females also increased, and decreased when body sizes decreased (Table 4; Figs. 4B and 4E). There were, however, a number of possible alternative situations, the most common of them that SSD increased when body size of both sexes decreased (Figs. 4A and 4C; Table 4). In all individual branches in which this happened females were larger than males, so the increase in SSD was due to a relative larger reduction in male size (Table 4). The inverse situation, i.e., a decrease in SSD when body size of both sexes increased, was much less frequent (Figs. 4C and 4F; Table 4). But again, in the single branch with a SSD decrease larger than 5% when both sexes increased in size, females were larger than males, so the reduction in SSD was due to a relative larger increase of the male body size (Table 4 and S3). When only change above 5% was considered, the most common situation was no change in either males or females (Table 4), but the general pattern did not change.

There were branches in which there was a significant change in SSD with an increase in body size of only one of the sexes. When males were the sex that changed, there was generally an increase in SSD (17 out of 24 branches, Table 4). When females were the only sex to significantly change their size, SSD increased when female size decreased (three branches), or decreased when female body size increased (also three branches, Table 4). There was only one branch in which SSD decreased when only female body size decreased, but four when only male body size decreased. In none of the branches SSD increased only due to an increase in female body size, while in 10 SSD increased with only

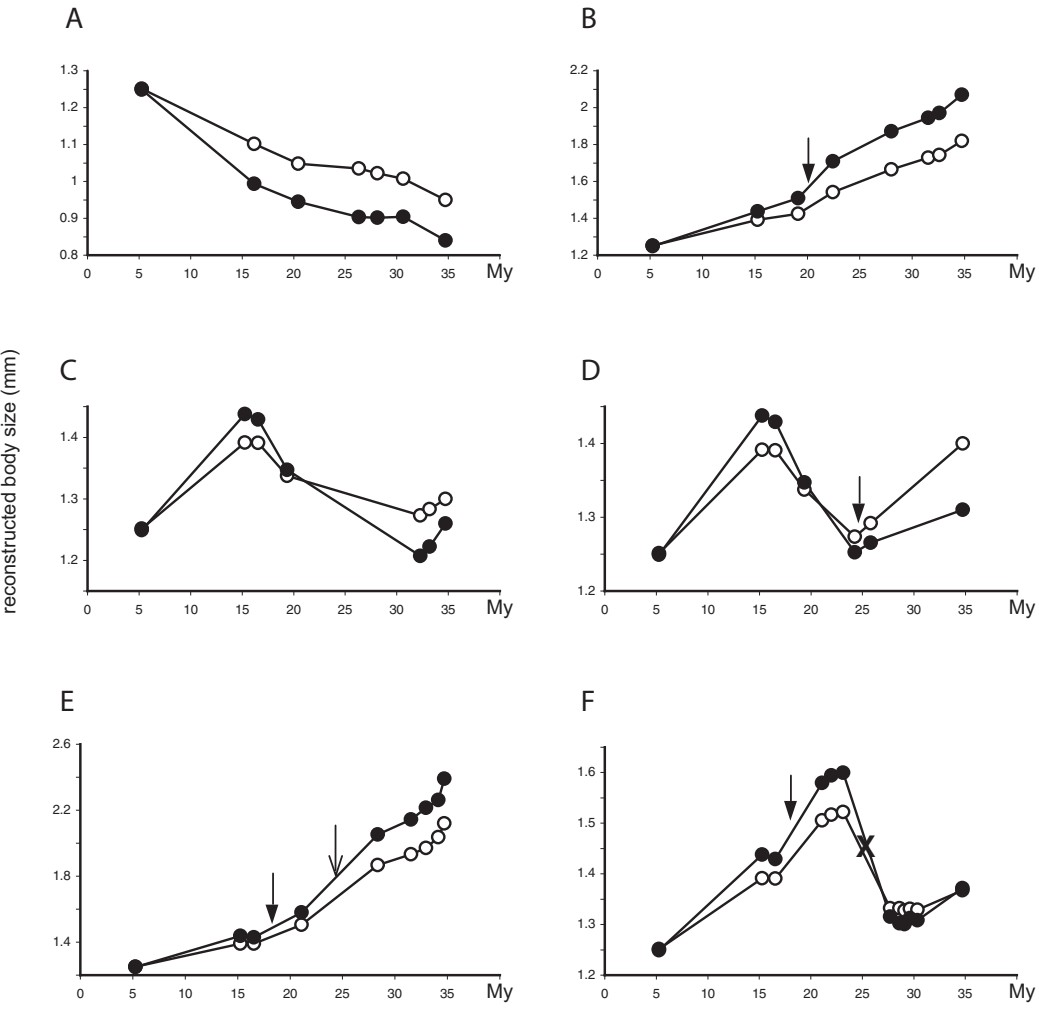

**Figure 4 Evolutionary trajectories of SSD in *Limnebius*.** Evolutionary trajectories of SSD in selected species of *Limnebius*. In the *y*-axis, reconstructed values of male (filled circles) and female (white circles) body sizes (mm); in the *x*-axis, nodes in the reconstructed evolution of the species (Figs. 2 and 5). Distances in the *x*-axis are proportional to time (My, note that the scale is reversed, i.e., the root of the Limnebiini tree is dated with time 0). Filled arrows, apparition of abdominal secondary sexual characters (SSC); empty arrows, apparition of SSC in the posterior tibiae; crosses, secondary lost of SSC, all according to the reconstruction in Fig. 5. (A) *Limnebius extraneus*; (B) *Limnebius papposus*; (C) *Limnebius arenicolus*; (D) *Limnebius fallaciosus*; (E) *Limnebius fretalis*; (F) *Limnebius nitidus* complex—note that this trajectory is the same for all the species in this complex, due to the short terminal branches and the uncertain relationships among the species.

an increase in male body size (Table 4). In all these cases, it can be considered that changes in SSD were not associated with an overall size increase, as change was restricted to only one sex. Overall, there were 24 branches in which SSD significantly changed only due to male change, while change was only due to females in seven branches (Table 4; $p < 0.005$ of equal probabilities assuming a binomial distribution). For all studied species, the reconstructed average size change along the evolutionary path was larger for males than females.
**Table 4 Changes in body size and in SSD in the individual branches of the phylogeny.** Summary changes in male (*lm*) and female (*lf*) body size and in SSD in the individual branches of the phylogeny (see Table S3 for quantitative values of all individual branches). In 'all changes,' all branches are coded either with positive or negative change, irrespective of the amount of change. In 'change >5%,' changes lower than 5% of the initial value in male or female body size, and changes in SSD lower than 5% of the total range of SSD changes, are coded as '=' (i.e., with no change).

| *lf* | *lm* | SSD | All changes | Change >5% |
|---|---|---|---|---|
| − | − | − | 25 | 8 |
| − | + | − | 5 | 0 |
| + | − | − | 7 | 0 |
| + | + | − | 10 | 1 |
| − | − | + | 31 | 9 |
| − | + | + | 6 | 0 |
| + | − | + | 8 | 2 |
| + | + | + | 38 | 16 |
| − | = | − | | 1 |
| = | − | − | | 4 |
| + | = | − | | 3 |
| = | + | − | | 3 |
| − | = | + | | 3 |
| = | − | + | | 7 |
| + | = | + | | 0 |
| = | + | + | | 10 |
| − | = | = | | 2 |
| − | − | = | | 9 |
| + | + | = | | 6 |
| = | + | = | | 3 |
| = | = | =(+/−) | | 43 |

**Table 5 Correlations between differences in rates of evolution between males and females and changes in each sex separately.** Correlation of the differences in the rates of evolution between males and females in all individual branches of the molecular phylogeny with the change in each sex separately, measured in darwins and in absolute phenotypic change.

| | | Darwins | | | | Absolute phenotypic change | | | |
|---|---|---|---|---|---|---|---|---|---|
| | | *m* | | *f* | | *m* | | *f* | |
| | n | $R^2$ | Slope | $R^2$ | Slope | $R^2$ | Slope | $R^2$ | Slope |
| *Limnebius* | 130 | 0.33*** | 0.34 | n.s. | −0.02 | 0.56*** | 0.39 | 0.11*** | 0.24 |
| *Limnebius* s.str. | 98 | 0.30*** | 0.31 | n.s. | −0.003 | 0.59*** | 0.37 | 0.17*** | 0.28 |
| *Bilimneus* | 28 | 0.87*** | 0.93 | n.s. | −0.45 | 0.78*** | 0.95 | 0.15* | −0.82 |

Notes:
* $p < 0.05$;
** $p < 0.01$;
*** $p < 0.001$.

Differences in the rate of phenotypic evolution of male and female body size in the individual branches, as measured in darwins, were positively correlated to the change in males, but negatively (albeit not significantly) with that of females (Table 5).

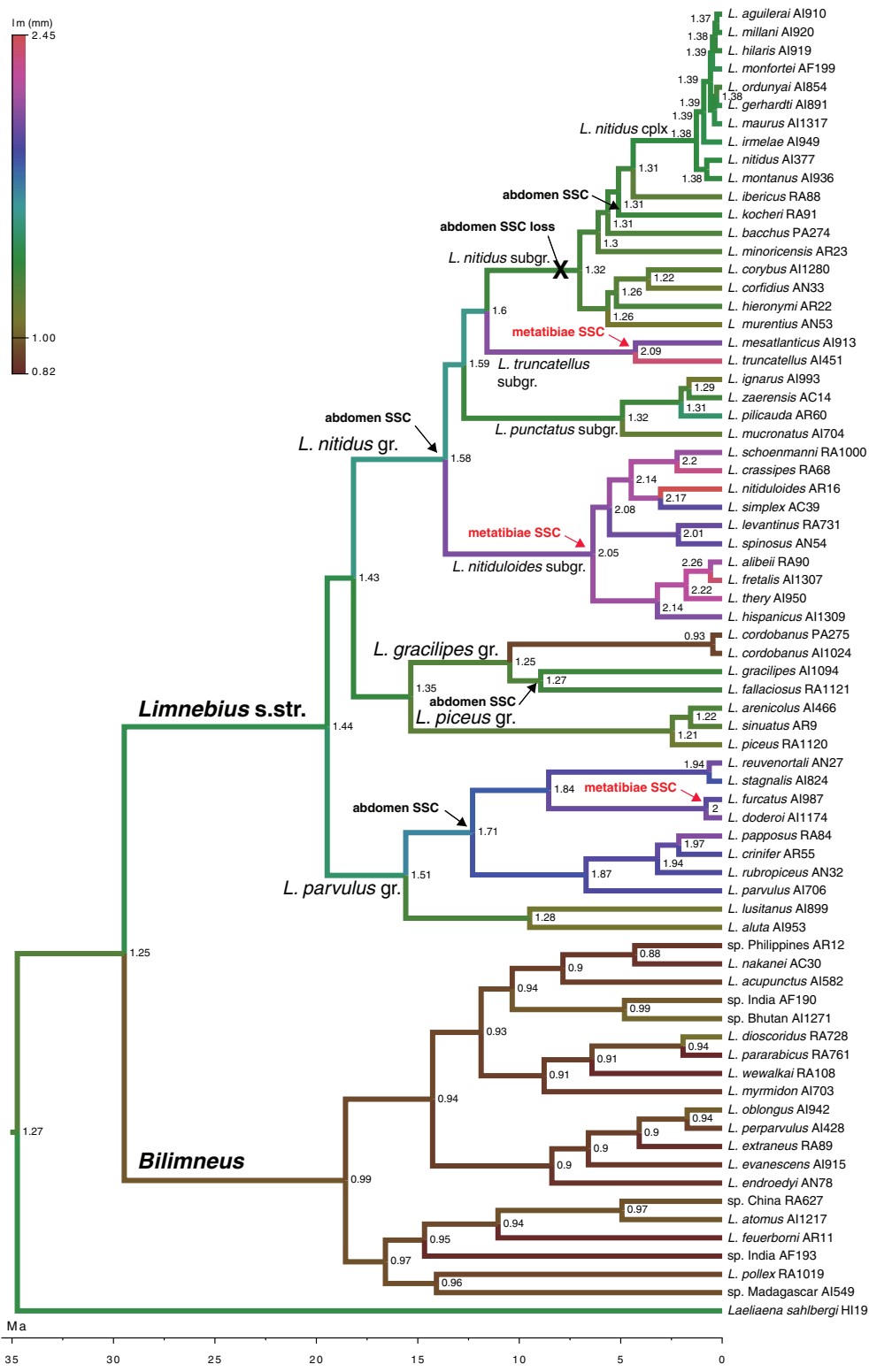

**Figure 5 Evolution of male body size in *Limnebius*.** Evolution of male body size (*lm*, mm) in the phylogeny of *Limnebius* as reconstructed in BEAST using a Brownian motion model of evolution. The origin and lost of secondary sexual characters (SSC) was reconstructed in MESQUITE using parsimony. Numbers in nodes, reconstructed value of male body size. Arrows mark the appearance of SSCs, crosses mark their loss.

When measured in absolute phenotypic change, differences between male and female body size were positively correlated with the change of both sexes, but with a stronger correlation and a steeper slope for males (Table 5).

## Evolution of secondary sexual characters

According to our reconstruction, modifications in the hind tibiae appeared three times independently in the phylogeny (Fig. 5). Within the *Limnebius nitidus* group, species in the *Limnebius nitiduloides* subgroup have a row of setae (Figs. 4E and S3), and two species, *Limnebius truncatellus* and *Limnebius mesatlanticus*, have the distal part of the male hind tibiae strongly widened (Fig. S2; *Jäch, 1993*). In addition to these, in three closely related species within the *Limnebius parvulus* group males have wider hind tibiae (*Limnebius doderoi*, *Limnebius furcatus*, and *Limnebius gridellii*, Fig. 5).

According to our reconstruction, the protuberance in the male abdomen appeared independently in the *Limnebius gracilipes* group (in the clade excluded *Limnebius cordobanus*, *Rudoy, Beutel & Ribera, 2016*) (Figs. 4D and S3) and in the *Limnebius nitidus* group. Within the later, it was secondarily lost in the *Limnebius nitidus* subgroup, with the exception of *Limnebius kocheri* (Figs. 4F and S3). The other modifications of the abdomen of males occur in two of the subgroups of the *Limnebius parvulus* group (Fig. 5). The two species of the *Limnebius setifer* subgroup have a medial impression, and the species of the *Limnebius parvulus* subgroup a tuff of setae, with the exception of *Limnebius glabriventris*, very close to *Limnebius parvulus*, which likely lost it secondarily (*Jäch, 1993*). There was no molecular data for the species of the *Limnebius setifer* subgroup (Fig. S3), so it remains uncertain whether there may have been a single origin for the secondary modifications of the abdomen, which subsequently diverged in the two subgroups, or they appeared independently.

# DISCUSSION

## Rensch's rule

Our results confirm the general validity of Rensch's rule in the genus *Limnebius*, that is, that body size of males is evolutionary more labile than that of females (*Rensch, 1950*; *Fairbairn, 1997*). There are several lines of evidence supporting this conclusion. First, the regression between male ($y$-axis) and female ($x$-axis) body size had a slope larger than one (i.e., a positive allometry). As a consequence, the ratio male/female body size (rSSD) was correlated mostly with male body size, while the correlation with female body size was lower and in some cases not significant, indicating that males drive the evolution of SSD. Also, when the evolution of SSD was reconstructed in the individual branches of the phylogeny evolutionary rates of male body size were generally higher than in females, and when females had higher rates there was a secondary reduction in SSD. When the change in SSD was measured in darwins (a compound measure including rate), it was also correlated to absolute changes in male, but not female body size, for which the correlations were negative but not significant. Results were very similar when regressions were obtained using raw data or phylogenetic independent contrasts, and for the later,

when using the molecular phylogeny (with a subset of the species) or the estimated phylogeny with all species.

The most commonly accepted cause for Rensch's rule is the continued action of directional sexual selection on the body size of males (*Abouheif & Fairbairn, 1997*; *Székely, Freckleton & Reynolds, 2004*). The increase in male body size resulting from sexual selection produces an increase in SSD when males are larger, but a decrease in SSD if females are larger. This model assumes a correlated evolution between male and female body size, so that when males increase in size females also increase, although at a lower rate (i.e., the correlation is <1) (*Maynard Smith, 1977*; *Fairbairn & Preziosi, 1994*). Our results are in general agreement with this model of the effect of sexual selection in combination with a general correlation between male and female body sizes, as changes in SSD were most frequently associated to changes in body size of both sexes (although larger in males). However, there were a variety of cases that did not fit the model, the most common being an increase in SSD while both sexes decrease in size. The contrary situation, with an increase in body size of both sexes leading to a decrease in SSD, was more infrequent, being found only in one branch within *Bilimneus*, the subgenus with females generally larger than males. Both cases contradict the association of SSD with an overall increase in the body size of both sexes resulting from the action of sexual selection on male body size, but still show larger changes in males than in females, in agreement with Rensch's rule.

There are two other possible cases in which there is no association between changes in SSD and changes in body size of both sexes. One is isometry, defined in our case as a change larger than 5% in body size of both sexes in parallel, but with a change in SSD of less than 5% (i.e., considered to be not significant). The reconstructed branches with isometric change in the *Limnebius* phylogeny occurred in lineages with small species with low SSD. The second case is an increase in SSD due only to an increase in male body size, with no change (or a change lower than 5%) in the females. This situation was more frequent in the phylogeny, suggesting that in some circumstances there may be a decoupling of the evolution of the male and female body size. It has been suggested that when body size is subjected to other selective forces females should approach their optimal size independently of the size of males, which may be mostly driven by sexual selection (*Lande, 1980*). The cases in which changes in SSD were associated to changes in only one sex challenge the assumption that selection on body size of one sex will always drive the evolution of the other due to their overall genetic correlation. Our results are in agreement with simulation studies showing that in species with large populations SSD evolves free of constraints, despite the genetic correlation between the sexes (*Reeve & Fairbairn, 2001*).

In general, although the overall evolution of SSD in *Limnebius* seems to conform to a standard model with sexual selection favouring an increase in male body size with female body size also increasing due to genetic correlation, there was considerable variation, with deviating patterns in some lineages. The high evolutionary lability of SSD was confirmed by its low phylogenetic signal as measured with the $K$ statistic, lower than for male and—especially—female body size.

## Relationship between SSD and aedeagus size and complexity

We found a general positive correlation between SSD and size and complexity of male genitalia, which would suggest that genital characters are also subjected to sexual selection, in parallel to male body size. However, in a previous study, *Rudoy & Ribera (2016)* did not found clear evidence for the presence of directional selection in the evolution of the complexity and size of the male genitalia in the genus *Limnebius*. Although the most complex genitalia are always present in the larger species, small species may also have complex genitalia. The size of the male genitalia was also evolutionary very labile, with no clear trends and a large variance, especially in *Limnebius* s.str., with the larger species and the more complex genitalia (*Rudoy & Ribera, 2016*). Similarly, we found here that lineages with uniformly complex genitalia may have a wide variation in SSD (as in e.g., the species of the *Limnebius punctatus* subgroup, Table S1), contrary to hypotheses linking Rensch's rule with the evolution of genital characters (*Bonduriansky & Day, 2003*). In other insects (e.g., water striders), a positive correlated evolution between non-intromittent genitalia and SSD has been reported, but there was no correlation between the shape of intromittent genital traits and SSD (*Arnqvist & Rowe, 2002*).

## Secondary sexual characters

Secondary sexual characters of the external morphology appeared several times independently in *Limnebius*, but generally in large species with high SSD. Male secondary sexual traits are often linked to directional sexual selection (*Petrie, 1988*; *Wilkinson, 1993*; *Simmons & Tomkins, 1996*; *Wilkinson & Taper, 1999*; *Simmons, 2013*; *Santos & Machado, 2016*), which will be supported by their association with species with high SSD. It is also interesting to note that the only loss of SSC affecting a relatively diverse lineage (within the *Limnebius nitidus* subgroup) occurred in a lineage that also secondarily reduced the SSD and the complexity of the male genitalia in some of the species, although other still have relatively complex aedeagus (*Rudoy, Beutel & Ribera, 2016*). Due to the uncertainty in the topology of the *Limnebius nitidus* subgroup (*Rudoy, Beutel & Ribera, 2016*), it is not possible to assess if the presence of protuberance in *Limnebius kocheri* is homologous to that of the other species of the group (i.e., it is sister to the rest of the species within the subgroup) or if it acquired the character independently, although the different conformation (short and acute in *Limnebius kocheri*, long, oblique and medially impressed in other species) suggest the later possibility (*Jäch, 1993*). This would also agree with an alternative topology grouping in a monophyletic clade all linages within the *Limnebius nitidus* group with SSC in the abdomen, in which case the absence of SSC in the *Limnebius nitidus* subgroup (with the exception of *Limnebius kocheri*) could be ancestral and not secondary (*Rudoy, Beutel & Ribera, 2016*).

The presence of SSC seems to be more linked to the complexity of the genitalia, both for the characters present in the terminal segments of the abdomen and the extreme modifications of the legs. According to our reconstruction, the modification of the posterior legs appeared after the development of SSC in the abdomen, and they are also not linked to SSD. Thus, they are conserved in species with a secondarily rSSD close to 1, as for example in some species of the *Limnebius nitiduloides* group, which has a strong

variation in SSD despite having uniformly complex genitalia (*Rudoy & Ribera, 2016*). As already noted, the relationship between SSC and complex genitalia is, however, not reciprocal, as there are groups with a complex genitalia but without SSC (as in e.g., the *Limnebius nitidus* complex or the *Limnebius punctatus* subgroup, Table S1). It must be noted that we only studied SSC in the external morphology of the male, but there is the possibility that there are other, less apparent SSC potentially subjected to sexual selection, such as e.g., chemical attractants, or even some behavioural traits.

## Concluding remarks

Our results demonstrate that the evolution of SSD dimorphism in the genus *Limnebius* was largely driven by changes in males, thus providing strong support for the prevalence of Rensch's rule. However, the increase in SSD was not always linked to an overall size increase in both sexes, and was not always associated to the presence of male SSC, contrary to the expectations under the hypothesis of sexual selection as the primary cause of Rensch's rule. Although most species with SSC had a strong SSD, with males larger than females, SSC are evolutionary more derived, appearing generally when species had already increased their size. In *Rudoy & Ribera (2016)* it was shown that differences in the evolution of the male genitalia between *Bilimneus* and *Limnebius* s.str. were largely due to an increase in the variance of the change in the later, in which males are generally larger than females and which includes the larger species and the species with the stronger SSD and the more complex genitalia. This raises the possibility that the primary driver for the evolution of male body size is simply their larger evolutionary variance, maybe related to the lack of constraints associated with egg development and reproduction likely acting on females. In *Limnebius* s.str., sexual selection, with the subsequent development of SSC, may have been triggered in lineages that already had larger males and complex genitalia, reinforcing these pre-existing traits. The stronger constraints in the variability of males in subgenus *Bilimneus* remains to be explained, but it may be related to unknown differences in mating behaviour or other traits related to reproduction.

## ACKNOWLEDGEMENTS

We are grateful to M.A. Jäch (NMW, Wien) and P. Perkins (MCZ, Harvard) for allowing us to study the collections of their institutions and for supporting this work in many ways. A. Rudoy thanks M.A. Jäch for his help during his stay in the NMW. We also thank the comments and advice of F. García-González (EBD, Seville), A. Cordero-Rivera (University of Vigo), E.S.A. Santos (Universidade de São Paulo) and an anonymous referee.

### Funding

This work was funded by a JAE PhD studentship (CSIC) to A. Rudoy, the Spanish Ministerio de Economía y Competitividad (projects CGL2010-15755 and CGL2013-48950-C2-1-P), a Salvador de Madariaga grant in the Phyletisches Museum

in Jena (PRX14/00583) to I. Ribera and the 'Secretaria d'Universitats i Recerca del Departament d'Economia i Coneixement de la Generalitat de Catalunya' (project SGR1532). There was no additional external funding received for this study. The funders had no role in study design, data collection and analysis, decision to publish, or preparation of the manuscript.

## Grant Disclosures

The following grant information was disclosed by the authors:
JAE PhD studentship (CSIC).
Spanish Ministerio de Economía y Competitividad: CGL2010-15755 and CGL2013-48950-C2-1-P.
Salvador de Madariaga grant: PRX14/00583.
Secretaria d'Universitats i Recerca del Departament d'Economia i Coneixement de la Generalitat de Catalunya: SGR1532.

## Competing Interests

The authors declare that they have no competing interests.

## Author Contributions

- Andrey Rudoy conceived and designed the experiments, performed the experiments, analysed the data, contributed reagents/materials/analysis tools, wrote the paper, prepared figures and/or tables, and reviewed drafts of the paper.
- Ignacio Ribera conceived and designed the experiments, performed the experiments, analyswed the data, contributed reagents/materials/analysis tools, wrote the paper, prepared figures and/or tables, and reviewed drafts of the paper.

## Data Deposition

   The raw data has been supplied as Supplementary Dataset Files.

## Supplemental Information

Supplemental information for this article can be found online at http://dx.doi.org/10.7717/peerj.3060#supplemental-information.

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
