# Peer review of "Evolution of sexual dimorphism and Rensch’s rule in the beetle genus Limnebius (Hydraenidae): is sexual selection opportunistic?"

_PeerJ, doi:10.7717/peerj.3060_

## Round 0.1 · original submission · Major Revisions

Both reviewers found great potential in your study, but raised a number of important issues that need to be addressed before the manuscript can be accepted.

Reviewer 1 ·

Basic reporting

Generally, the presentation of the data and results should be improved. Importantly, the orthography and syntax should be revised throughout the manuscript. Some abbreviations are unnecessary and confusing (lm, lf, per, fd). I suggest to remove them with the exception of “SSD”. Also, axis labels are essential in data presentation but are lacking throughout which is very inconvenient (or at least they did not show in my copy).

Experimental design

The authors use a large data set of a beetle genus which is especially well suited to study SSD because SSD varies qualitatively among species. Investigating the relationship of SSD with genital size and complexity is an exciting aspect of this manuscript addressing the potential role of sexual selection on males driving Rensch’s rule.

I am however worried about the correct identification of females. As in any comparative study, correct identification is essential. I think the authors should outline and explain their approach more thoroughly.

It is great that the authors discuss the potential of non-representative estimates into account. Do the results change if the analyses are repeated only with species that have sufficient data?

I do not get the difference between the relative perimeter and the fractal dimension method. Are fractal dimensions corrected for size as well? If not, its correlation with SSD is difficult to interpret.

Validity of the findings

Major comments:
I am not convinced that the three lines of evidence outlined in the discussion are independent. Analyzing the relationship between male and female size (1) is equivalent to a correlation of rSSD (the use of this index is problematic, see below) with male or female size (2). Using these two analyses as independent lines of evidence is thus flawed. Regressing rSSD against sex-specific body size is not necessary and suffers from statistical drawbacks (given that the y-axis is a ratio). A similar reasoning applies to the quantification of correlated increase or decrease of size and SSD within a branch in the phylogenetic tree (lines 216-223). This is very similar to what independent contrast do (if not identical). While the incorporation of a 5% threshold is a valuable tool to account for measurement errors I think that this cannot be used as independent evidence. I am not convinced that these analyses give more information than a classic plot of log male size vs. log female size using independent contrasts. I see clear evidence for Rensch’s rule and the first analysis suffices.

l.233: For conceptual reasons, regression models should be forced through the origin if the relationship between independent contrasts are investigated. Was this done here? The figures indicate otherwise. RMA slopes should then become less steep but are likely to remain significantly larger than unity.

l.254: It has been known for a while in the SSD literature that simple ratios of male and female size are problematic (here the case with rSSD), especially if SSD ranges from female- to male-biased. I suggest to stick to standard indices which have less statistical issues (either SDI by Lovich&Gibbons (1992) or log (m/f), both discussed in Fairbairn et al. 2007: sex, size, gender roles: chapter 1).

Generally, the notion that sexual selection acts opportunistically is not clearly outlined. I think this aspect should receive more attention in the discussion. Also, the lack of association of SSD variation with secondary sexual traits may not be as robust. Secondary sexual traits are very diverse in its nature and not limited to stark variation in morphology. Sexual dimorphisms in other traits (eg. coloration, chemical ornaments, etc) are not investigated here. The limitations of using few but very divergent traits could be more appreciated.

Minor comments:
I am not proficient in reconstructing phylogenies and can thus not evaluate its quality.

The authors discuss that their use of BM may not necessarily be the best model. Were alternative models of trait evolution considered?

Was aedeagus size standardized for overall body size? If not the correlation between genital size and SSD could just represent Rensch’s rule itself.

l.392: I do not understand the conclusion that genitalia are under sexual selection if they correlate to SSD.

Additional comments

The strength of this manuscript lies in its great data set which is highly suited to study Rensch's rule and the evolution of SSD. Although I disagree with some of the methods applied here and urge the authors to revise the grammar of the text, I like the manuscript and think that it furthers our understanding of SSD and its evolution.

·

Basic reporting

Comments on manuscript entitled "Evolution of sexual dimorphism and Rensch's rule in the beetle genus Limnebius (Hydraenidae) - is sexual selection opportunistic?”

Eduardo S.A. Santos (I sign all reviews)

I have now finished reviewing the manuscript and will present my comments following PeerJ’s format below. These comments are intended for both the authors of the manuscript and the editor. I hope that these comments help the authors to revise the manuscript in order to make it a better publication.

1. Basic reporting
Clear, unambiguous, professional English language used throughout?
Concerning the use of language, the manuscript is generally well written.

Intro & background to show context. Literature well referenced & relevant?
Overall the literature is well used throughout the manuscript.

L47-53: What do you mean by “population fertility”? The view expressed in this sentence gives the impression that you are equivocally looking at the population as the selective unit, whereby males will be selected to be small and females to be large in order to minimize the population resource expenditure. Please rewrite. The sentence that starts in L51 also does not make much sense, by mentioning that a species has achieved an “optimal” state of small male size.

L60-62: You need to better explain the logic of the “social reason” argument. Please cite a work that shows this line of evidence.

L64-67: I don’t follow the logic of this paragraph in this sentence. The paragraph started with an opening sentence about selection on body size. But here in L65-67 you are talking about sexual selection acting through female mate choice on other characters, such as the genitalia. It makes no sense to think about SSD in genital characters. Given that in the next paragraph (L70-72) you talk about diversity of male genitalia, you need to make the argument in the current paragraph very clear, with a clear development and a concluding remark that allows readers to understand how genitalia links into the story.

L77-80: The hypothesis that sexual selection is indeed one of the mechanisms that explains the Rensch’s rule pattern is supported by the work on wing pigmentation, because wing pigmentation is “fairly" unrelated to other forms of selection (natural, fecundity, etc). The same cannot be said about body size or other body size related structures. I believe that this logic needs to be clearly stated in order for readers to understand why the wing pigmentation study makes sense.

L86-90: This sentence is not clear. Please rewrite, especially the first part. There is a typo in the beginning, and the statement is hard to understand, given that you are talking about allometry, a negative relationship between males and females needs to be better explained. Which sex is in the x-axis, which is in the y?

L90-93: Here Santos & Machado (2016) is a study that demonstrates Rensch’s rule in a trait other than body size.

L94-100: What are your hypotheses? This paragraph provides only a general idea of what your objective is, but I cannot understand what the predictions will be. Especially when it comes to the last sentence of the paragraph, about the correlation between SSD and other traits. What are the hypotheses here? What is the argument that leads to a positive (or negative) correlation hypothesis between SSD and genitalia, for instance? This is very important for the rest of the study.

L110-114: Which secondary sexual characters? Why is this a suitable system for the study of the origin of SSD? This is not clear from the description of the subgenera.

Structure conforms to PeerJ standard, discipline norm, or improved for clarity?
The structure of the manuscript, with regards to the sections presented, conforms to PeerJ’s standard and is clear.

Figures are relevant, high quality, well labelled & described?
All figure captions would benefit from a thorough revision. I found them to be poorly explained and not self contained, i.e. one needs to refer back to the text several times to understand the content and context of the figures.

Figure 1 needs better explanation about what is in each panel, axes need to have titles. I don’t understand what are the units of the data. The data are reported as “0,03”, for example, so you need to replace all commas with full stops to separate the decimals.

Figure 2: This figure needs a better scale legend. You need to inform us where the 1.0 is in the colour scale, what a value > 1 means and a < 1 means.

Figure 3: Please add titles to the axes.

Figure 4: Please add titles to the axes.

Figure 5: This figure needs a better scale legend. You need to inform us where the 1.0 is in the colour scale, what a value > 1 means and a < 1 means.

Raw data is supplied?
No. I did not have access to the raw data of the study (it was not among the files downloaded with the reviewer pack). The authors provide a supplemental table containing the summary of the morphological data used. It would be interesting to provide a working data pack with the phylogeny used.

Experimental design

2. Experimental design
Original primary research within Scope of the journal?
Yes, the research presented in the manuscript is within PeerJ’s scope.

Research question well defined, relevant & meaningful. It is stated how research fills an identified knowledge gap.
The research question is not clearly stated. As I mentioned earlier L94-100, without clear hypotheses, readers will not understand some of the methodological decisions that the authors undertook. Moreover, without clear hypotheses, it is very hard to understand the logic of why one would investigate the presence or not of certain patterns.

Rigorous investigation performed to a high technical & ethical standard?
See my comment about L118-128.

Methods described with sufficient detail & information to replicate?
No. There are several places that deserve more attention from the authors when it comes to detail and information about the methods. I cannot find an explanation of how SSD was calculated. There are several ways to calculate SSD. Without a clear explanation in the methods, there simply is not a way to understand what the SSD values mean. What does a 1.24 SSD value mean? This is crucial as the whole manuscript hinges on such interpretation. The explanation of SSD appears completely out of context in the result section (L254). Please correct this issue.

Validity of the findings

3. Validity of the findings
Impact and novelty not assessed. Negative/inconclusive results accepted. Meaningful replication encouraged where rationale & benefit to literature is clearly stated.
I can see that the manuscript has potential to be beneficial to the literature, however in its current form it has just not reached that mark. The introduction needs to be better written in order to provide the study’s hypotheses/predictions so that readers can understand the rationale. The methodology needs to address some issues regarding detail (that will surely occur if the paper has clear hypotheses).

Data is robust, statistically sound, & controlled.
There seems to be problems with the raw data, as I discuss below in the comment about L118-128. Please see that comment for further details. If the authors do not remove the ambiguous data from the dataset, I am not certain that the findings can be unequivocally supported.

Conclusion well stated, linked to original research question & limited to supporting results.
The studies conclusion is linked to the research conducted. I will once again say that without clearly stated hypothesis, the conclusion is not well stated.

Additional comments

4. General comments

L46-47: Change “males, which mostly provide just genetic information” to “males that mostly provide only genetic information"

L75: Replace “where males” with “in which males”. Same for the females in the same sentence. “Where” denotes a physical locality; “in which” refers to all other forms of association.

L103-104: Replace “with the only exception…” with “with the only exception being the absence from saline habitats”.

L118-128: This is the most critical part of the manuscript. The idea of studying Rensch’s rule and the evolution of SSD in a globally distributed and closely related group of animals is interesting. Nevertheless, the whole study hinges on good identification of the specimens to their respective species. From what is being said in the first paragraph of the Methods section (L118-128), there are several identification issues that could lead to mistakes in the estimation of SSD. I appreciate that the authors are honest and transparent about the problems, but if there are issues in identification of males and females to their correct species, these need to be removed from the analyses. The authors have mentioned that several species coexist and are troublesome to be attributed to a species. They use as a form of identification for females the association with males. The first issue here is that what the authors mean by association is not clear. Secondly, even if we are talking about copulation association, there is the very likely possibility that males are copulating with closely related females from another species, thereby invalidating the method used to attribute females to their respective species. Out of the 86 species for which you have female data, how many of these suffer from the issues highlighted above? My suggestion is that you remove all the species for which there are issues in identifying individuals to the correct species. Otherwise, I (and all the readers for that matter) will not be convinced by the robustness of the data.

L136-154: Why did you quantify genitalia morphology? This relates back to the introduction. Without clear hypotheses I cannot understand why you measured this variables. The same comment applies to the body size variables. You need to state what your predictions are here in the methods in order for readers to understand why certain variables were quantified. You need to justify why you measured the length of the aedeagus and also the perimeter? What do these variables represent? Why are you interested in the correlation between SSD and the “complexity” of the variables? What does the “fractal dimension” of the aedeagus represent biologically? Please clarify all these questions.

L196-205: More information needs to be provided in order for readers to understand what you mean by these secondary sexually selected traits. In what context are they used? Why did you measure them? Why reconstruct their origin? Since no hypotheses are provided, it is impossible to understand why you are interested in estimating these traits.

L339: the RMA regression does not estimate a correlation.

L339-342: This could be very likely due to the small sample size of Bilimneus. An explanation that must be considered.

L353-366: This paragraph starts with an opening sentence that leads readers to believe that you will create an argument about sexual selection being a mechanism, but during the development of the paragraph, you do not talk about sexual selection anymore, and the concluding statement does not supply a conclusion as to what can be interpreted in regards to sexual selection. Please address this issue.

L383-388: I do not agree with the conclusion that sexual selection is the main mechanism explaining the evolution of SSD. From your findings, it is simply not possible to make this inference. At least more information on the role of body size would be needed in order to go in this direction. Does body size in males influence mating success? What other mechanisms could account for directional selection on body size? How can one make the inference that faster rates of body size evolution in males is caused by sexual selection? Your results, while interesting, highlight a pattern, i.e., faster evolution of male body size. The interpretation of what this means depends on other data, such as the role of body size in male-male contests, female mate choice, etc.

L391-393: I don’t understand the logic behind this statement. Why would a correlation between SSD and complexity of male genitalia mean that genital traits are also under sexual selection. Please clarify.

L391-404: You need to explain the logic here to your readers. Citing Bonduriansky & Day and Arnqvist & Rowe does not explain the the rationale of why one would expect a correlation between body size SSD and genitalia complexity. This needs to be clear from the introduction.

L407-L432: Once again, the whole discussion of secondary sexual traits is based around correlations of these traits with SSD and with the complexity of the genitalia. You do not delve into what are the predictions, nor what does it mean for these traits to be correlated with other traits. I once again emphasise the need of clear hypotheses based on theoretical predictions. Otherwise, your discussion offers very little insight from the results you obtained.

L437-440: This is the first time in the manuscript that you state — even though not clearly — a prediction from a hypothesis. Please rethink how you have written the introduction in order to build the hypothesis from the theoretical expectations about sexual selection leading to Rensch’s rule.

L633: Table 1: The is a mistake in Table 1, the row that contains the results of the Bilimneus molecular analysis has a p-value of “< 0.05”, but the 95% CI overlaps with 1, meaning that the slope is not statistically different from 1.

---

## Round 0.2 · Minor Revisions

As you'll notice from the reviews, both reviewers recognise that the manuscript has improved considerably, but they also indicate a few minor modifications that need to be addressed before the study is acceptable for publication.

Reviewer 1 ·

Basic reporting

As prior to the revision, the manuscript is appropriately embedded in the current literature. The presentation of data has been greatly improved.

Experimental design

no comment

Validity of the findings

As mentioned earlier, I cannot evaluate the validity of any phylogenetic reconstruction.

Additional comments

I think that the authors did a good job in revising this manuscript.
The only comment I do have is again related to the "lines of evidence"-argument (I do apologize that I am such a pain).
I agree that the authors do not explicitly sell the results of their different methodologies as "independent lines of evidence". However, reading the lines 344 to 352, the impression that they do, does persists (I believe that the impression of the reader is in this case more important than the intentions of the authors. After all, it is my job to point that out). Although, this may only be an issue of phrasing and maybe not so much a conceptual issue.
Let's have a look at this section (excerpt line 346 to 350):
"(1) the regression between male (y-axis) and female (x-axis) body size had a slope larger than one (i.e. a positive allometry); (2) the ratio male/female body size (rSSD) was correlated mostly with male body size, while the correlation with female body size was lower and in some cases not significant, indicating that males drive the evolution of SSD;..."

In this case, the second finding is given by the first one simply due to the mathematical relationship (and because the same data are used, at least to my reasoning). By stressing both findings equally (in this case by numbering them), and not mentioning that the second analysis is redundant, I get the impression that the same mathematical relationship is used twice to support the same argument. This is clearly not done intentionally but may nevertheless elicit suspicion by certain readers. If this does not concern the authors (which I hope it does), I do not object their decision to leave it as it is.

·

Basic reporting

There are still some small issues with the language. See my comments below.

The literature is properly cited and used, and the data is now shared as supplemental files.

Experimental design

The authors addressed comments that both myself and Reviewer #1 made about the research question that helped to make the interpretation of the study's design easier. The methods are now sufficiently described and more clear than in the original version of the manuscript.

Validity of the findings

With the changes made to the manuscript, I am now comfortable with the validity of the findings and the conclusions provided by the authors.

Additional comments

Comments on MS #14209v2

L53-56: I had previously inquired about the use of the term “population fertility” (see my original comment L47-53), to which you replied that you had removed this sentence from the revised version of the manuscript. However, upon reading the revised version, I found that the sentence is still there, with a slight change in how the term is used. I still believe that the use of the term “fertility of a population” is equivocal. I think that you can remove the passage “the fertility of the population depends more on females than on males,” and use only the rest of the sentence to make your argument about the evolution of body size in males and females.

L67-69: Thank you for the explanation about what you meant by “social reason”. It is now clear to the reader.

L72: There is a comma after “body, but also other…”. Moreover, change “used for male-male” to “used in male-male”.

L73-75: Rewrite as “Similarly, in the latter case, in addition to male body size, other characters may be involved, especially genital characters…”

L78: Change “a number of” with “several”.

L94-97: Thank you for addressing this bit, it is much clearer now.

L103-106: Thank you for clarifying this bit.

L112: “In this work, …”

L134: “As a study group, …”

L153-165: Thank you for rewriting this section, it reads much better now.

L191: “In most analyses, …”

L192: Ditto

L215-219: In your response letter, you explained very well what the fractal dimension represents biologically. However, you did not change the text in this method section. I would strongly recommend that the authors change the text here to make it clearer to readers that are not familiar with the approach. It is not enough to ask readers to refer to Rudoy et al. 2016 for further details. Most readers will not do this and at this point you have lost the interest of your audience.

L231: “For our analyses, …”

L264: “Due to these limitations, …”

L266: “secondary loss”

Other than these comments, I am happy with the changes that the authors have made to the manuscript and believe that it will be an interesting contribution to the field of the evolution of sexual size dimorphism.

---

## Round 0.3 · accepted · Accept

I'm happy with how you addressed this last round of suggestions, and I believe it is ready for publication.